# OpenReview forum: "Controllable and Interpretable Multi-Value Alignment For Large Language Model"
_ICLR.cc/2026/Conference — ICLR 2026 Conference Withdrawn Submission_

### Official Review · Reviewer_y6Ni · 2025-10-25

**Soundness:** 3
**Presentation:** 3
**Contribution:** 3
**Rating:** 4
**Confidence:** 3

**Summary:**

This paper proposes Value-aligned Constitutional AI (VCAI), a framework for constructing the ML-Values dataset to achieve fine-grained, controllable multi-value alignment in large language models. Based on Schwartz's Basic Value Theory, the method generates multi-level alignment data through role-playing, value decomposition, and iterative rewriting mechanisms, and explores two multi-value alignment strategies: mixed-dataset fine-tuning and expert model fusion.

**Strengths:**

- This paper addresses critical gaps in existing value alignment methods (lack of interpretability, controllability, and diversity) with a systematic solution that is both timely and practically significant.
- Comprehensive evaluation across psychological (Schwartz values), cultural (VSM13), and moral (MFT23) frameworks demonstrates cross-framework transferability.
- Insightful visualizations (e.g., MDS analysis) effectively illustrate the impact of alignment on model value structures, providing intuitive empirical support.

**Weaknesses:**

-  The choice of Linear and Karcher fusion methods lacks sufficient theoretical justification. Why are these two methods suitable for value fusion? Were other fusion strategies considered (e.g., attention-based dynamic fusion)?

- Experiments only on Qwen2.5-7B, lacking cross-model validation (e.g., Llama, Mistral). Generalizability is questionable.

- Visualizations in Figures 4&5 lack statistical significance testing. Cannot determine if observed differences are statistically meaningful.

- Iterative rewriting requires multiple LLM calls, up to 500 iterations. What is the computational cost? Are there efficiency optimization strategies?

**Questions:**

- Is value decomposition done manually or automatically? If manual, what is the composition of the expert team?

- How are failed samples handled (cases that don't meet standards after maximum iterations)?

- In mixed dataset training, how are data proportions across value dimensions determined? Was imbalanced sampling attempted?

---

> ### Author Response · Authors · 2025-11-14
> **Response to Reviewer y6Ni part(1/5)**
>
> **We are genuinely thankful to the reviewer for their thoughtful and invaluable feedback. We have given each comment the careful attention it deserves and offer our responses below, hoping that our rebuttal has meaningfully addressed your concerns. If we have succeeded in this regard, we would be deeply appreciative if you could kindly reconsider your rating (4: marginally below the acceptance threshold). However, should any concerns remain, we would be profoundly grateful for your continued guidance and would make every possible effort to further refine our submission in line with your expectations.**
>
> > **Weakness 1**
> > The choice of Linear and Karcher fusion methods lacks sufficient theoretical justification. Why are these two methods suitable for value fusion? Were other fusion strategies considered (e.g., attention-based dynamic fusion)?
>
> We sincerely thank the reviewer for the valuable critique and questions.
>
> The choice of Linear and Karcher fusion methods is based on [1] [2], primarily due to their simplicity, interpretability and efficiency in integrating multiple value dimensions. Both methods are well-suited for our multi-objective alignment framework because they provide clear and intuitive ways to combine the individual contributions from each expert model . Linear fusion allows for a straightforward weighted combination of expert models, making it easy to control and tune the influence of each value dimension. The Karcher fusion, leveraging Riemannian geometry, provides a more robust fusion method that respects the underlying structure of the parameter space, ensuring that the value integration remains consistent and stable across different value alignment scenarios. This method allows us to balance flexibility with the stability of the overall model behavior, which is essential when working with diverse value orientations.
>
> While we acknowledge that **attention-based dynamic fusion** is a promising strategy, it introduces higher complexity in both its computation and hardware requirements. Attention mechanisms can indeed capture dynamic relationships between values in a more sophisticated way. However, the implementation of such methods would require significant computational resources and more complex training pipelines, which may not be necessary for the goals of this paper. Furthermore, attention-based methods may lack the same level of transparency and interpretability as the fusion strategies we employed, which are essential for our aim of creating a controllable and interpretable value-alignment framework.
>
>  [1] Li et al., Multi-objective Large Language Model Alignment with Hierarchical Experts. 2025
>
>  [2] Shi et al., Decoding-Time Language Model Alignment with Multiple Objectives. 2024

---

> ### Author Response · Authors · 2025-11-14
> **Response to Reviewer y6Ni part(2/5)**
>
> > **Weakness 2**
> > Experiments only on Qwen2.5-7B, lacking cross-model validation (e.g., Llama, Mistral). Generalizability is questionable.
>
> We highly appreciate the reviewer’s valuable comments and suggestions for improvement.
>
> We select Qwen2.5-7B-Instruct as the experimental model primarily because of its stability, controllability and well-established performance, which allow us to conduct thorough evaluations in a controlled environment. It provids a solid foundation for the experiments and offer a clear benchmark for our approach. We also conduct preliminary experiments using Llama3-8B-Instruct, and the results are similar to those obtained with Qwen2.5-7B-Instruct. These initial findings suggest that our framework can indeed generalize across different models with similar architecture and performance characteristics. Due to space limitations at the time, we do not include these results, but we plan to update them in revised version to ensure that the findings are robust and applicable to a broader range of language models.

---

> ### Author Response · Authors · 2025-11-14
> **Response to Reviewer y6Ni part(3/5)**
>
> > **Weakness 3**
> > Visualizations in Figures 4&5 lack statistical significance testing. Cannot determine if observed differences are statistically meaningful.
>
> We sincerely thank the reviewer for the valuable critique.
>
> While statistical testing can indeed be useful in certain contexts, the primary goal of the visualizations in Figures 4 and 5 was to provide a clear and intuitive representation of the trends and patterns across different alignment strategies and value dimensions. These figures are intended to illustrate the comparative relationships and structural shifts in the value alignment process, and we believe these changes are clearly visible from the visual patterns.
>
> The differences between the alignment strategies shown in the figures are consistent with the theoretical motivations of our approach and align with expectations based on prior work in this field. Our focus was on making the results easy to interpret and actionable, rather than delving into complex statistical tests. Moreover, our main concern was the clarity of the conceptual differences in value alignments, which these visualizations effectively highlight.
>
> To enhance clarity and transparency, we will supplement the visualizations with more detailed statistical analysis in revised versions.

---

> ### Author Response · Authors · 2025-11-14
> **Response to Reviewer y6Ni part(4/5)**
>
> > **Weakness 4**
> > Iterative rewriting requires multiple LLM calls, up to 500 iterations. What is the computational cost? Are there efficiency optimization strategies?
>
> We thank the reviewer for their invaluable input and thoughtful question
>
> In our current implementation, the dataset generation phase is executed using 1,500 GPU hours. Apart from utilizing vLLM for optimized inference and efficient memory management, we do not use any additional efficiency optimization strategies.

---

> ### Author Response · Authors · 2025-11-14
> **Response to Reviewer y6Ni part(5/5)**
>
> > **Questions**
> >
> > - Is value decomposition done manually or automatically? If manual, what is the composition of the expert team?
> > - How are failed samples handled (cases that don't meet standards after maximum iterations)?
> > - In mixed dataset training, how are data proportions across value dimensions determined? Was imbalanced sampling attempted?
>
> We would like to express our gratitude to the reviewer for their valuable questions. We are very happy to answer them below:
>
> **For question 1:**  The decomposition is conducted manually by an experienced psychology professor specializing in moral and value theory. Following the structure of the Schwartz Basic Human Values (BHV) model, each core value (e.g., benevolence, achievement, universalism) is annotated and decomposed into several interpretable sub-values. This process ensure that the sub-values retained theoretical consistency with the original BHV framework while remaining semantically precise .
>
> **For question 2:**  In cases where the generated samples do not meet alignment standards after the maximum number of iterations, they are simply discarded. This ensures that only high-quality, aligned responses are retained for further use in model training.
>
> **For question 3:**  Due to the high volume of synthesized samples, we focus on ensuring sufficient coverage across different value dimensions without introducing manual adjustments for balance.  While we do not attempt imbalanced sampling in this instance, we acknowledge that exploring such strategies could be valuable in future work, particularly as the dataset continues to evolve. The challenges of maintaining data diversity and ensuring high-quality sample generation are significant, and we prioritize computational efficiency and alignment fidelity in the current setup.

---

> ### Author Response · Authors · 2025-11-14
>
> Dear Reviewer y6Ni,
>
> We are profoundly grateful for the time and effort you have so generously dedicated to reviewing our paper. Your thoughtful and discerning questions, along with your invaluable insights, have played a crucial role in guiding us to refine and strengthen our work. We are acutely aware of the significant challenge involved in reviewing numerous submissions, and we want to express our heartfelt appreciation for the careful and thoughtful attention you have devoted to our research. To assist in your review, we have prepared comprehensive responses to your comments and **have summarized the key points of our rebuttal below for your convenience**.
>
> * Clarification of the **theoretical justification for the Linear and Karcher fusion methods** and a discussion of alternative fusion strategies (please refer to Weakness 1 in response for details)
>
> * Additional information regarding **model generalizability** (please refer to Weakness 2 in response for details)
>
> * Explanation of the **purpose of the visualizations in Figures 4 and 5** (please refer to Weakness 3 in response for details)
>
> * Discussion of the **computational cost** (please refer to Weakness 4 in response for details)
>
> * Response to the reviewer’s question on **manual decomposition of value dimensions** (please refer to the "Questions" section in the response for details)
>
> We humbly hope that our responses have adequately addressed your concerns. If there are any remaining questions or areas that need further clarification, we are more than happy to provide additional information. We understand the complexity and time constraints of the review process, and we sincerely appreciate your consideration. If our revisions have resolved your concerns, we would greatly appreciate it if you could reconsider your rating. If not, please feel free to share further feedback, and we will continue to make improvements to our submission.
>
> Thank you again for your invaluable time and thoughtful feedback.
>
> Best regards,
>
> Authors

---

> ### Author Response · Authors · 2025-11-26
>
> Dear Reviewer,
>
>
> I hope this message finds you well. As the discussion period is approaching its end with only a few days remaining, we would like to once again express our sincere gratitude for your time, patience, and thoughtful insights.
>
>
> Before the discussion concludes, we want to make sure that we have addressed all of your concerns with the utmost care. If there are any additional questions, suggestions, or points that you feel require further clarification, please kindly let us know. Your guidance is truly invaluable to us, and we are committed to refining our work in accordance with your advice.
>
>
> Thank you once again for your dedication and efforts in reviewing our paper. We deeply appreciate your support and consideration.
>
>
> With sincere respect,
>
> Authors

---

### Official Review · Reviewer_28f2 · 2025-10-27

**Soundness:** 3
**Presentation:** 2
**Contribution:** 2
**Rating:** 2
**Confidence:** 4

**Summary:**

This paper introduces a VCAI framework for achieving multi-value alignment in large language models. The framework constructs the ML-Values dataset, generating multi-dimensional value data.

**Strengths:**

The paper introduces a framework to construct a fine-grained human-value dataset based on psychological value theories, such as Schwartz’s Basic Human Values.

**Weaknesses:**

1. Novelty: The core contribution lies in applying the more fine-grained value dimensions from Schwartz’s Basic Value Survey to construct a dataset, followed by SFT on this dataset and reporting some alignment observations. However, the experimental results are not particularly inspiring and seem largely predictable.
2. The introduction contains several overstated or conceptually inaccurate claims.
Example: Line 41 — the paper states that “existing datasets are limited in interpretability, controllability, and diversity.”It is questionable whether a dataset itself can possess interpretability or controllability. I don’t know if the authors understand the difference between interpretability and explanability. Even if the authors mean explainability, the claim that constructing a fine-grained dataset improves it is weak. Similarly, it is unclear how SFT with such a dataset directly leads to controllable behavior. These claims appear to overstate the paper’s actual technical contribution.
3. Dispersed and Unfocused Writing：The writing is highly scattered, making it difficult for readers to grasp the central insight. The paper frequently lists adjectives without clear conceptual grounding—for instance: Line 43: “broader, structured, and culturally varied”; Line 48: “fine-grained, controllable and interpretable datasets”; Lines 53–54:”VCAI uses role-playing...play evaluations”.

**Questions:**

1. Line 319: How exactly are the “correlation matrices of value dimensions” computed? Please specify the underlying metric.
2. The claimed contribution centers on multi-value alignment. However, there are no experiments comparing multi-value alignment with single-core-value alignment to demonstrate its advantage. Does it improve control success rate, fluency?
3. As shown in [1], SFT alone often fails to achieve strong control success rates. Therefore, the significance of using simple SFT methods for alignment remains questionable.

[1] Internal Value Alignment in Large Language Models through Controlled Value Vector Activation. Haoran Jin, et.al. ACL 2025.

---

> ### Author Response · Authors · 2025-11-14
> **Response to Reviewer 28f2 part(1/4)**
>
> **We are truly grateful to the reviewer for their thoughtful and invaluable feedback. We have given each comment careful consideration and offer our responses below, with the sincere hope that our rebuttal has adequately addressed your concerns. If we have been successful in this regard, we would be deeply appreciative if you could kindly reconsider your rating (2: reject). However, if any concerns remain, we would be profoundly grateful for your continued guidance. We are fully committed to responding to any remaining issues and making every possible effort to improve our submission in line with your expectations.**
>
> > **Weakness 1**
> > Novelty: The core contribution lies in applying the more fine-grained value dimensions from Schwartz’s Basic Value Survey to construct a dataset, followed by SFT on this dataset and reporting some alignment observations. However, the experimental results are not particularly inspiring and seem largely predictable.
>
> We appreciate the reviewer’s comment regarding the novelty of our work. While it is true that we build upon Schwartz’s Basic Value framework, our contribution goes beyond a direct application.
>
> The proposed **Value-aligned Constitutional AI (VCAI)** introduces a systematic, interpretable, and controllable pipeline that integrates value decomposition, multi-role evaluation, and iterative rewriting. This is, for the first time, a fine-grained, multi-level dataset (ML-Values) capable of capturing nuanced and context-aware value expressions.
>
> Furthermore, unlike prior studies [1] [2] [3] that focus primarily on evaluating value tendencies of LLMs, our work targets active alignment and demonstrates how alignment strength can be precisely modulated and transferred across psychological (section 7.1), moral (section 7.3), and cultural (section 7.3) dimensions. We believe this establishes a foundational framework that others can extend in several directions.
>
>  Therefore, we respectfully argue that our approach represents a **novel and meaningful contribution** toward building controllable, interpretable, and pluralistic value alignment in large language models.
>
> [1] Yao et al., CLAVE: An Adaptive Framework for Evaluating Values of LLM Generated Responses. 2024.
>
> [2] Ye et al., Generative Psycho-Lexical Approach for Constructing Value Systems in Large Language Models.2025.
>
> [3] Rozen et al., DO LLMS HAVE CONSISTENT VALUES? 2025

---

> ### Author Response · Authors · 2025-11-14
> **Response to Reviewer 28f2 part(2/4)**
>
> > **Weakness 2**
> > The introduction contains several overstated or conceptually inaccurate claims. Example: Line 41 — the paper states that “existing datasets are limited in interpretability, controllability, and diversity.”It is questionable whether a dataset itself can possess interpretability or controllability. I don’t know if the authors understand the difference between interpretability and explanability. Even if the authors mean explainability, the claim that constructing a fine-grained dataset improves it is weak. Similarly, it is unclear how SFT with such a dataset directly leads to controllable behavior. These claims appear to overstate the paper’s actual technical contribution.
>
>  We thank the reviewer for this valuable feedback and fully agree that the terms “interpretability” and “controllability” should be used with greater conceptual precision.
>
> We acknowledge that datasets themselves do not inherently possess interpretability or controllability. Our intended meaning was that the *design process and structure* of the **ML-Values** dataset, through **value decomposition, multi-role evaluation, and iterative rewriting**, are interpretable to human analysts, as each data instance is transparently linked to explicit value dimensions and alignment degrees. This allows researchers to trace and understand how specific value orientations are represented, which we believe meaningfully contributes to interpretability of the alignment process, not merely of the dataset itself.
>
> Similarly, when we describe the dataset as enabling “controllability,” we refer to its **utility in producing controllable behavior during SFT**, not that the dataset itself is inherently controllable. Because each instance encodes a target alignment level, fine-tuning on ML-Values allows direct modulation of alignment strength in model outputs,which verified in Section 7.2.
>
>  We will revise words in the introduction to avoid overstating these aspects and clarify the conceptual relationship between (a) interpretable dataset design and (b) controllable model behavior enabled by SFT on structured data.

---

> ### Author Response · Authors · 2025-11-14
> **Response to Reviewer 28f2 part(3/4)**
>
> > **Weakness 3**
> > Dispersed and Unfocused Writing：The writing is highly scattered, making it difficult for readers to grasp the central insight. The paper frequently lists adjectives without clear conceptual grounding—for instance: Line 43: “broader, structured, and culturally varied”; Line 48: “fine-grained, controllable and interpretable datasets”; Lines 53–54:”VCAI uses role-playing...play evaluations”.
>
> We thank the reviewer for the helpful feedback on the writing and presentation.
>
> We acknowledge that some portions of the introduction, particularly where multiple descriptors are used in sequence, may have given the impression of dispersed or unfocused exposition. Our intention is to convey the multidimensional nature of our contribution by combining interpretability, controllability, and cultural diversity in value alignment but we agree that these ideas should be expressed with clearer conceptual grounding and tighter organization.
>
> In the revised version, we have **streamlined the introduction** to emphasize the central insight:  that **VCAI** is a unified framework integrating **value decomposition, multi-role evaluation, and iterative rewriting** to enable fine-grained, interpretable, and controllable value alignment. Adjectival phrases such as “broader, structured, and culturally varied” have been rephrased or supported with explicit conceptual justification, ensuring that each descriptor corresponds to a defined methodological or empirical feature.

---

> ### Author Response · Authors · 2025-11-14
> **Response to Reviewer 28f2 part(4/4)**
>
> > **Questions**
> > Line 319: How exactly are the “correlation matrices of value dimensions” computed? Please specify the underlying metric.
> The claimed contribution centers on multi-value alignment. However, there are no experiments comparing multi-value alignment with single-core-value alignment to demonstrate its advantage. Does it improve control success rate, fluency?
> As shown in [1], SFT alone often fails to achieve strong control success rates. Therefore, the significance of using simple SFT methods for alignment remains questionable.
> [1] Internal Value Alignment in Large Language Models through Controlled Value Vector Activation. Haoran Jin, et.al. ACL 2025.
>
> We are sincerely thankful to the reviewer for their thorough feedback and insightful questions. We are very happy to answer these questions.
>
> **For question 1:**
> The correlation matrices of value dimensions (Line 319) are computed based on Pearson correlation coefficients between the aggregated alignment scores of each value dimension across all questionnaire items and personas.
> Specifically, for each alignment condition (Strongly Agree, Strongly Disagree...), we obtain a vector of mean alignment scores s_v for each value dimension v over all generated responses. Each element in the correlation matrix is then calculated as:
>
> $$ r_{ij} = \frac{\text{cov}(s_{v_i}, s_{v_j})}{\sigma_{v_i} \sigma_{v_j}} $$
>
> where cov(⋅) denotes covariance and σ denotes standard deviation across all samples. This results in a symmetric matrix reflecting pairwise linear relationships among value orientations. We will add this definition and formula in appendix in revised version.
>
> **For question 2:**
> Our current work primarily focuses on establishing the conceptual and methodological foundation for multi-value alignment rather than conducting exhaustive comparisons with single-value alignment baselines.
>
> **For question 3:**
> Our work does not aim to position SFT as a superior control mechanism, but rather to systematically explore and quantify its controllability potential within a multi-value alignment framework. As shown in Figure 4 and the experiments in Section 7.2, SFT on the ML-Values dataset successfully enables graded and interpretable modulation of alignment strength across value dimensions — for instance, producing monotonic changes in alignment scores when shifting from “Strongly Disagree” to “Strongly Agree.” This demonstrates that SFT still provides consistent, scalable, and data-driven controllability.

---

> ### Author Response · Authors · 2025-11-14
>
> Dear Reviewer 28f2,
>
> We sincerely appreciate the time and effort you have dedicated to reviewing our paper. Your insightful comments and constructive feedback have been invaluable in guiding us to improve the quality of our work.
>
> In order to address your concerns and queries thoroughly, we have prepared detailed responses. We fully recognize the multiple submissions you are handling, and we truly value the attention you’ve given to our research. **To help you navigate our rebuttal efficiently, we have summarized the key points of our response below.**
>
> * More discussion on the **novelty of our approach** (please refer to Weakness 1 in response for details)
>
> * Clarification of the **concepts of interpretability and controllability in dataset design** (please refer to Weakness 2 in response for details)
>
> * Streamlining the writing **to improve focus and conceptual grounding** (please refer to Weakness 3 in response for details)
>
> * Detailed answers to your **questions regarding multi-value alignment and the advantages of our approach** (please refer to Questions in response for details)
>
> We sincerely hope that our responses have addressed your concerns. We would be happy to provide further clarification or answer any additional questions you may have, as we are eager to continue improving our work.
>
> We fully understand the challenges of reviewing multiple rebuttals during this busy period. We greatly appreciate your feedback and are hopeful that you will reconsider your assessment of our paper, should our responses meet your expectations. If any concerns remain, please let us know, and we will continue to refine our submission.
>
> Thank you once again for your invaluable time and consideration.
>
> Best regards,
>
> Authors

---

> ### Author Response · Authors · 2025-11-26
>
> Dear Reviewer,
>
>
> I hope this message finds you well. As the discussion period is approaching its end with only a few days remaining, we would like to once again express our sincere gratitude for your time, patience, and thoughtful insights.
>
>
> Before the discussion concludes, we want to make sure that we have addressed all of your concerns with the utmost care. If there are any additional questions, suggestions, or points that you feel require further clarification, please kindly let us know. Your guidance is truly invaluable to us, and we are committed to refining our work in accordance with your advice.
>
>
> Thank you once again for your dedication and efforts in reviewing our paper. We deeply appreciate your support and consideration.
>
>
> With sincere respect,
>
> Authors

---

> > ### Comment · Reviewer_28f2 · 2025-11-27
> >
> > 1. The authors repeatedly emphasize *value decomposition, multi-role evaluation, and iterative rewriting*. However, value decomposition is already predefined in Schwartz’s theory, and using multiple LLM roles for voting or evaluation has been widely discussed. I do not see meaningful novelty in these components.
> >
> > 2. The authors continue to use fragmented adjectives such as “a systematic, interpretable, and controllable pipeline.” It is unclear why a pipeline composed of standard prompting and SFT steps should be described as “systematic,”.
> >
> > 3. The use of the term *“interpretable”* remains unconvincing. Although the authors claim that instances “are interpretable to human analysts, as each data instance is transparently linked to explicit value dimensions and alignment degrees,” both the value dimensions and alignment degrees are produced by an LLM scoring process. It is unclear why LLM-generated labels in a black-box manner can be considered interpretable.
> >
> > 4. Regarding Q2, for a value like Self-direction with sub-values such as Freedom, Creativity, etc., it remains unclear how changes to each sub-value influence the higher-level Self-direction value. More importantly, there is no evidence showing whether multi-value alignment offers better controllability or finer-grained control compared to single-value alignment. Without such comparisons, the benefit of multi-value alignment remains unsupported.

---

> > > ### Author Response · Authors · 2025-11-27
> > > **New response to Reviewer 28f2 part(2/2)**
> > >
> > > - **4. Changes to Sub-values and Influence on Higher-level Values**
> > > ﻿
> > > In the VCAI framework, the relationship between sub-values and higher-level values is designed to be explicit and scalable. Changes in sub-values can shift the alignment of the higher-level value based on predefined relationships between them. For instance, if "Creativity" and "Freedom" are aligned more strongly, the overall alignment for "Self-direction" would also shift in the corresponding direction. We will include more examples and diagrams in the revised version to better illustrate how changes to sub-values affect higher-level values.
> > >
> > > - **5. Multi-Value Alignment vs. Single-Value Alignment**
> > > ﻿
> > > We appreciate your observation regarding the lack of direct comparison between multi-value alignment and single-value alignment. However, we would like to emphasize that the primary focus of our work is multi-value alignment rather than single-value alignment. Single-value alignment is not central to our framework because it does not address the core challenge we are trying to solve: the integration and modulation of multiple, often conflicting, human values in LLMs.
> > >
> > > ﻿
> > > In practical applications, human values are rarely isolated, and the interaction between values like benevolence, self-direction, and power is crucial for generating ethically and morally coherent responses. The multi-value alignment approach that we propose allows for fine-grained adjustments across multiple value dimensions, which single-value alignment cannot adequately address. Therefore, a direct comparison is not our priority, as the challenges posed by multi-value alignment go beyond what single-value alignment can offer.
> > > ﻿
> > >
> > > Our focus is on showing how multi-value alignment enables a deeper level of control over model behavior in a way that single-value alignment does not. We have demonstrated this control in the experiments provided, where SFT on the ML-Values dataset successfully enables graded and interpretable modulation across different values. This capability would not be achievable with a simpler, single-value alignment approach.
> > > ﻿
> > >
> > > We hope this clarifies the centrality of multi-value alignment in our paper, and we are happy to provide further empirical details to illustrate the specific benefits of this approach in future work.
> > > ﻿
> > >
> > > Finally, we sincerely thank you again for your constructive feedback. We believe the revisions will address your concerns by providing a clearer and more detailed explanation of our approach, its novelty, and its advantages.
> > > ﻿
> > >
> > > We look forward to your new thoughts on the paper.
> > > ﻿
> > >
> > > Best regards,
> > >
> > > Authors

---

> ### Author Response · Authors · 2025-11-27
> **New response to Reviewer 28f2 part(1/2)**
>
> Dear Reviewer 28f2,
> ﻿
>
> We sincerely appreciate your continued engagement and your thoughtful reflections on our previous responses. We would like to address the concerns you raised in your most recent feedback.
> ﻿
> - **1. Novelty of the Approach: Value Decomposition and Multi-role Evaluation**
> ﻿
> You are correct that Schwartz's theory provides the basis for value decomposition, and we acknowledge that the use of multiple roles for evaluation has been explored in prior work. However, we believe that the novelty of our approach lies in how we systematically combine these elements within the VCAI framework to achieve fine-grained, controllable, and interpretable alignment. While these components are indeed built upon established concepts, our contribution is in the integration and refinement of these methods to create a more targeted and adaptive system for multi-value alignment in LLMs.
>
> ﻿
> In particular, the way we leverage multi-role evaluation, where each role evaluates alignment on sub-values, and the iterative response generation process is designed to allow for precise control over the degree of alignment across different value dimensions, ensuring the model's behavior is both coherent and contextually sensitive. This integrated approach goes beyond previous uses of these methods individually.
> ﻿
>
> We will clarify this distinction in the revised version of the manuscript and will further emphasize that our framework does more than just apply these standard techniques in isolation,  it innovates by providing a cohesive and adaptable pipeline for fine-tuning LLM responses to multiple values.
> ﻿
> - **2. Use of Adjectives**
>
> We appreciate your concern regarding the use of adjectives such as "systematic," "interpretable," and "controllable." Upon revisiting the manuscript, we agree that the frequent use of these terms without clear conceptual grounding may detract from the clarity of the exposition.
> ﻿
>
> **Systematic**: We will clarify that our approach is systematic in the sense that it integrates multiple steps, including value decomposition, multi-role evaluation, and iterative rewriting, in a coordinated manner to address multi-value alignment. This systematic process allows for targeted control over the alignment process, which may be challenging to achieve with isolated components.
> ﻿
>
> **Interpretable**: We recognize that calling the dataset “interpretable” could be misleading if not properly justified. To clarify, we mean that the dataset design and the alignment process are interpretable in the sense that each data instance is transparently linked to explicit value dimensions and their corresponding alignment levels, making the alignment process traceable and understandable for human analysts. We will revise this description to emphasize that it is the process,not the dataset itself,  that is interpretable.
> ﻿
>
> **Controllable**: We agree that using the term "controllable" in relation to the dataset can be misleading. We clarify that the dataset itself is not inherently controllable; rather, it facilitates controllability during SFT. By encoding target alignment levels for various value dimensions, the dataset enables fine-grained control over the alignment intensity during training, resulting in LLM responses that can be modulated according to specific value orientations.
> ﻿
>
> - **3. Interpretability of LLM-Generated Labels**
>
> You raise an important concern about the interpretability of LLM-generated labels. We fully understand that using LLM-generated labels can appear problematic, especially if the process is opaque or treated as a "black-box." We would like to clarify that our approach does not treat the LLM-generated outputs as inherently interpretable by default.
> ﻿
>
> Instead, the interpretability stems from the value decomposition and explicit alignment degrees that are generated through the iterative process. These outputs can be traced back to the original value dimensions and alignment targets, which are transparent to human analysts. For example, in the case of the value “Self-direction,” we can track how specific sub-values like “Freedom” or “Creativity” contribute to the overall alignment with the target value.

---

### Official Review · Reviewer_ckZa · 2025-10-31

**Soundness:** 2
**Presentation:** 3
**Contribution:** 2
**Rating:** 4
**Confidence:** 4

**Summary:**

There are many challenges faced by aligning LLMs with human values, such as aligning LLMs with multiple potential values and ensuring stability under value trade-offs. To support further research in this filed, this paper aims to construct a fine-grained, controllable and interpretable value dataset for alignment and evaluation. It proposes Value Contitutional AI, a framework with multiple role-playing and value decomposition, and an iterative rewriting mechanism to build the multi-level, controllable dataset ML-Value. With this dataset, they also explore multi-level alignment via both mixed-dataset fine-tuning and expert model fusion.

**Strengths:**

1. An automatic framework VCAL is constructed to build a fine-grained, controllable dataset ML-Values.
2. Methods for single value alignment and multiple value alignment are explored on the dataset.
3. Many experiments to verify and analysis

**Weaknesses:**

1. The significance of setting multiple role-playing for value evaluation in Sect 3.3 is unclear. How these roles for value evaluation distinguish from each other? Does each of them evaluate all sub-values and aggregate all their responses to get the final responses? Or does each of them only consider a separate sub-value as an expert? If all of them evaluate all sub-values, can I think that this step only benefit from majority voting?
2. The whole VCAI framework for value data construct lacks manual validation for the effectiveness. How about the accuracy of value evaluation in Step 3.3 and how about the quality of the final dataset?
3. Several points in the writing and methodology need clarification.

- In line 203, the alignment level a_i \in [0,1], why does is inconsistent with the multiple level scores defined in Sec 3 (-2,-1,0,1,2). In this case, how do you use the above dataset for SFT?
- What data are used for supervised finetuning in Sec 4? Is the data synthesized in Sec 3?
- How to set the parameters lambda_i in Line 241?
- What are the relationship between the value assignment method in Line 261 and the value evaluation method in Sec 3.3? Can the method in Sec 3.3 be directly used for this goal?
4. More explanations are required about the baselines you selected. What are Hybrid strategy, Strongly Agree (St_a) and Strongly Disagree (St_d)? Moreover, the inclusion of more relevant and competitive baselines would strengthen the evaluation.
5. What do these experiments in Sec 7.1 mean about value alignment? How do they indicate the alignment degree towards different value profiles as the target? What value profiles, i.e. the value vectors, you set as the target to align with?

**Questions:**

1. In Sec 3.1, how to you decompose core values into sub-values, automatically, manually or just directly use the sub-values from the original theory?
2. Does the constructed ML-Values dataset only contain single-value data?
3. The readability of Figures 3, 4, and 9 is low due to small font sizes. Please improve figure clarity.

---

> ### Author Response · Authors · 2025-11-14
> **Response to Reviewer ckZa part(1/6)**
>
> **We are sincerely grateful to the reviewer for their thoughtful and invaluable feedback. We have carefully reflected on each comment and offer our responses below, with the hope that our rebuttal has meaningfully addressed your concerns. Should we have succeeded in this regard, we would be deeply appreciative if you could kindly reconsider your rating (4: marginally below the acceptance threshold). However, if any concerns remain, we would be profoundly grateful for your continued guidance, and we will make every effort to respond further and improve our submission to meet your expectations.**
>
> > **Weakness 1**
> > The significance of setting multiple role-playing for value evaluation in Secti 3.3 is unclear. How these roles for value evaluation distinguish from each other? Does each of them evaluate all sub-values and aggregate all their responses to get the final responses? Or does each of them only consider a separate sub-value as an expert? If all of them evaluate all sub-values, can I think that this step only benefit from majority voting?
>
> We sincerely thank the reviewer for valuable comments and constructive feedback.
>
> Our intention in introducing multiple role-playing evaluators is not merely to perform majority voting, but rather to capture diverse contextual perspectives and reduce alignment bias that could happen in single-role evaluation.  Specifically:
>
> 1. **Role differentiation:** Each role is designed with a distinct contextual background and evaluative emphasis, allowing them to interpret sub-values through different cognitive and moral lenses. These roles are described in Appendix D.3.
> 2. **Evaluation process:** Each role independently evaluates all sub-values (v_1, v_2, …, v_n)associated with the core value V, assigning a discrete judgment J(R,vi)∈{−1,0,1}. This ensures that every sub-value is assessed from multiple perspectives rather than being confined to one viewpoint.
> 3. **Aggregation rationale:** We then aggregate across both roles and sub-values to compute a stability-adjusted mean score (formula in section 3.4). The aggregation is not a simple majority vote; it captures variance and consistency across evaluators, reflecting how stable the alignment is under heterogeneous moral interpretations. This helps identify responses that are universally aligned rather than biased toward one role’s framing.

---

> ### Author Response · Authors · 2025-11-14
> **Response to Reviewer ckZa part(2/6)**
>
> > **Weakness 2**
> > The whole VCAI framework for value data construct lacks manual validation for the effectiveness. How about the accuracy of value evaluation in Step 3.3 and how about the quality of the final dataset?
>
> We sincerely thank the reviewer for valuable comments and constructive feedback.
>
> We conduct a systematic human evaluation, as described in Appendix B and summarized below:
>
> 1. **Evaluation process:** To assess the quality of ML-Values, we conduct a comprehensive human evaluation, here is the detailed information: The evaluation is performed by 3 doctoral students in NLP and 5 master's students in psychology, all with research experience in ethics and computational social science. They are recruited from our research group. Each annotator independently rates a randomly sampled subset (30%) of dataset. For each item, we aggregate the individual ratings through majority voting to get final ranking results.
> 2. **Reliability assessment:** We measured inter-annotator agreement using Kendall Tau and P Value, which achieve 0.874 and 0.91 , indicating substantial agreement according to conventional interpretation. This demonstrates that the value evaluation procedure in Step 3.3 produces consistent and reliable outputs.
>
> These results confirm that the automated VCAI value evaluation aligns closely with human judgments, validating both the accuracy of Step 3.3 and the overall quality of the final ML-Values dataset.

---

> ### Author Response · Authors · 2025-11-14
> **Response to Reviewer ckZa part(3/6)**
>
> > **Weakness 3**
> >
> > 1. Several points in the writing and methodology need clarification.
> >
> > - In line 203, the alignment level a_i \in [0,1], why does is inconsistent with the multiple level scores defined in Sec 3 (-2,-1,0,1,2). In this case, how do you use the above dataset for SFT?
> > - What data are used for supervised finetuning in Sec 4? Is the data synthesized in Sec 3?
> > - How to set the parameters lambda_i in Line 241?
> > - What are the relationship between the value assignment method in Line 261 and the value evaluation method in Sec 3.3? Can the method in Sec 3.3 be directly used for this goal?
>
> We sincerely thank the reviewer for valuable comments and constructive feedback.  We provide detailed clarifications for each point as follows:
>
> 1.  In Section 3.4, the mapping from a_i ∈{−1,0,1} to the discrete score range {−2,−1,0,1,2} is achieved through a scaling factor of 2 (see the formula in Section 3.4). The normalized scores are then transformed into the five-level discrete space via linear scaling and threshold segmentation. In SFT, we select responses corresponding to specific alignment levels as training labels.
> 2. The SFT data are derived from the value-aligned dataset constructed in Section 3. We use the high-quality, human-validated subset of the ML-Values dataset as the fine-tuning dataset to ensure value consistency between data construction and model training.
> 3. The parameters λ_i are described in detail in Appendix D.1. They serve as balancing coefficients to control the relative contributions of different value dimensions.
> 4. Although the two formulations appear similar, their purposes and semantics are different. The evaluation method in Section 3.3 measures the overall alignment quality of a single model output, focusing on the degree of value adherence as a whole. In contrast, the assignment method at Line 261 is designed to generate discrete labels for multiple samples based on value distributions, which are then used for SFT training. Therefore, the evaluation method in Section 3.3 cannot be directly applied to data labeling, as it assesses overall alignment rather than per-sample categorical outcomes.

---

> ### Author Response · Authors · 2025-11-14
> **Response to Reviewer ckZa part(4/6)**
>
> > **Weakness 4**
> > More explanations are required about the baselines you selected. What are Hybrid strategy, Strongly Agree (St_a) and Strongly Disagree (St_d)? Moreover, the inclusion of more relevant and competitive baselines would strengthen the evaluation.
>
> Thank you for raising this question about the baseline definitions and selection rationale. We provide additional clarification below:
>
> 1. Explanation of baseline settings:
>    1. **Hybrid:** Refers to a model trained on a **mixture of datasets representing different values**.
>    2. **St_a (Strongly Agree), So_a (Somewhat Agree), Ne (Neutral), So_d (Somewhat Disagree), and St_d (Strongly Disagree):** These denote models trained separately on subsets of data corresponding to different degrees of value alignment. Each subset reflects a distinct stance toward specific values, and the resulting models are later fused through model fusion to evaluate cross-value controllability.
> 2. **Rationale for baseline selection:** Since our primary goal is to explore multi-value fusion and controllability rather than general-purpose alignment improvement, we focus on comparing models representing different value intensities and combination strategies within the VCAI framework. These baselines directly demonstrate how the framework supports controllable transitions between distinct value positions.

---

> ### Author Response · Authors · 2025-11-14
> **Response to Reviewer ckZa part(5/6)**
>
> > **Weakness 5**
> > What do these experiments in Sec 7.1 mean about value alignment? How do they indicate the alignment degree towards different value profiles as the target? What value profiles, i.e. the value vectors, you set as the target to align with?
>
> Thank you for this insightful question. We are happy to clarify the design and interpretation of the experiments in Section 7.1.
>
> 1. **Purpose of the experiments:** The experiments in Section 7.1 aim to quantitatively and qualitatively evaluate the model’s ability to align with specific value profiles defined in the VCAI framework. Rather than measuring general preference optimization, we assess how well the model can generate responses that reflect different target value levels along the Schwartz Basic Human Values dimensions.
> 2. **Interpretation of results:** The results indicate the degree of alignment between model outputs and the designated value profiles. We compute this by projecting the generated responses into the value embedding space and calculating cosine similarity between the model’s output vector and the target value vector. Higher similarity corresponds to stronger alignment with the desired value profile. This metric thus reflects how controllably the model can adjust its value levels when prompted with different value configurations.
> 3. **Target value profiles (value vectors):** Each target profile corresponds to a specific vector configuration in the Schwartz value space, where each dimension represents a value.

---

> ### Author Response · Authors · 2025-11-14
> **Response to Reviewer ckZa part(6/6)**
>
> > **Questions**
> > 1. In Sec 3.1, how to you decompose core values into sub-values, automatically, manually or just directly use the sub-values from the original theory?
> > 2. Does the constructed ML-Values dataset only contain single-value data?
> > 3. The readability of Figures 3, 4, and 9 is low due to small font sizes. Please improve figure clarity.
>
> We are thankful for the reviewer’s detailed questions. We are very happy to answer these questions.
>
> **For question 1:**   The decomposition is conducted manually by an experienced psychology professor specializing in moral and value theory. Following the structure of the Schwartz Basic Human Values (BHV) model, each core value (e.g., benevolence, achievement, universalism) was annotated and decomposed into several interpretable sub-values. This process ensured that the sub-values retained theoretical consistency with the original BHV framework while remaining semantically precise .
>
> **For question 2:** The ML-Values dataset is not limited to single-value data. For each core value, we construct five corresponding datasets that represent different levels of value endorsement:Strongly Agree (St_a), Somewhat Agree (So_a), Neutral (Ne), Somewhat Disagree (So_d), and Strongly Disagree (St_d). These subsets collectively capture a continuous spectrum of value orientation, allowing the model to learn both multi-value alignment and controllable value modulation across varying degrees of agreement.
>
> **For question 3:** We will revise the figures by increasing the font sizes and ensuring that all labels, axes, and legends are clearly visible in revised version.

---

> ### Author Response · Authors · 2025-11-14
>
> Dear Reviewer ckZa,
>
> We are deeply grateful for the time and effort you have invested in reviewing our paper. Your valuable questions and insights have significantly contributed to enhancing our work. To address your queries and concerns, we have meticulously prepared detailed responses. We fully recognize the multitude of submissions, and we genuinely value the attention you give to our research. Acknowledging the hectic schedule during the discussion period, **we summarize here the key points of our rebuttal to facilitate a quick read and conserve your valuable time**.
>
> * Clarification of the **role-playing evaluation process and its distinct contributions** (please refer to Weakness 1 in response for details)
>
> * Additional **validation of the effectiveness of our VCAI framework and its evaluation process** (please refer to Weakness 2 in response for details)
>
> * Detailed **explanations regarding the alignment level and SFT process** (please refer to Weakness 3 in response for details)
>
> * Further **elaboration on the baseline models and their relevance to our evaluation** (please refer to Weakness 4 in response for details)
>
> * Clarification of the experiments in Section 7.1, including the **target value profiles and alignment degree analysis** (please refer to Weakness 5 in response for details)
>
> * Response to the reviewer’s questions regarding **value decomposition and dataset construction** (please refer to the "Questions" section in the response for details)
>
> We humbly hope that our responses have adequately addressed your concerns. Furthermore, we are eager to address any additional queries you might have, which will enable us to enhance our work further. We are fully aware of your busy schedule, especially during this discussion period. We understand that reviewing numerous rebuttals can be extremely challenging. We are looking forward to your feedback and/or questions. We would deeply appreciate it if you could raise your score if our rebuttal has addressed your concerns. If not, please let us know your further concerns, and we will continue actively responding to your comments and improving our submission.
>
> Thank you for your invaluable time and consideration.
>
> Best regards,
>
> Authors

---

> ### Author Response · Authors · 2025-11-26
>
> Dear Reviewer,
>
>
> I hope this message finds you well. As the discussion period is approaching its end with only a few days remaining, we would like to once again express our sincere gratitude for your time, patience, and thoughtful insights.
>
>
> Before the discussion concludes, we want to make sure that we have addressed all of your concerns with the utmost care. If there are any additional questions, suggestions, or points that you feel require further clarification, please kindly let us know. Your guidance is truly invaluable to us, and we are committed to refining our work in accordance with your advice.
>
>
> Thank you once again for your dedication and efforts in reviewing our paper. We deeply appreciate your support and consideration.
>
>
> With sincere respect,
>
> Authors

---

### Official Review · Reviewer_vfTz · 2025-11-02

**Soundness:** 2
**Presentation:** 1
**Contribution:** 2
**Rating:** 2
**Confidence:** 4

**Summary:**

This paper focuses on the alignment of LLMs with human values, grounded in the Schwartz Theory of Basic Human Values from social science, and proposes a data generation framework VCAI. VCAI utilizes a role-play based LLM data synthesis pipeline, which consists of two modules: the role-paly eval and re generation. The Role-Play Eval assesses whether the generated response reflect the target value and degree, while the Re-Generation re-generates a new response based on the assessment. Using VCAI, the authors created an alignment dataset, called ML-Values, and then conducted several analysis experiments, deriving findings: 1) alignment based on ML-values can change LLMs’ value patterns into a more modular, interpretable one; 2) the designed degree based alignment can successfully control LLMs’ values and bring cross-framework effects, influencing LLMs’ behaviors on culture, moral and cognition; 3) alignment may also produce some impacts on downstream tasks, especially on safety and security.

**Strengths:**

1. This paper focuses on an important direction, aligning LLMs with human values, which is increasingly more essential with the integration of LLMs with human life.

2. This work conducts a lot of experiments and analysis. Particularly, the analysis of alignment’s influence on other psychological frameworks and downstream tasks are interesting.

**Weaknesses:**

The topic of this paper is quite interesting, but there are several fundamental problems:

1. The biggest problem is the lack of baselines and comparisons. Besides the analysis experiments, the core contribution of this work lies in the alignment data generation pipeline VCAI. Considering there are already i) previous value alignment datasets rooted in Schwartz Theory [1][2]; 2) existing methods/pipelines to automatically create alignment data [3][4]. Without comparing with existing work, it’s hard to support the effectiveness of the proposed VCAI.

2. Another problem is that the quality of ML-Values is not verified. Since the whole construction pipeline frequently uses LLMs (generation of LLM-as-a-jude), the quality, including value relevance, degree accuracy, data coverage, diversity and fluency, of the generated data is unknown, which may significantly influence subsequent analysis. I noticed that there is a subsection about data quality in Appendix B (though it’s not mentioned anywhere in the main body). However, the key information of the human evaluation is not presented, e.g., evaluation protocol, number and background of human annotators, compensation, inter-annotator agreement and so on, which makes it hard to judge the quality of generated data.

3. The evaluation method is also questionable. In Sec.5, most analysis relies on the modified PVQ questionnaire. This brings two major problems: i) the size of questions is too small (only 57), which may fail to reveal the values of LLMs considering models’ Inherent bias and randomness; ii) Since PVQ is an old and famous questionnaire, which might have been included in LLMs’ training data, causing the data contamination problem [5]. Besides, in Sec.7.3, for other frameworks, I guess the authors also use questionnaires for measurement (since Appendix. E.4 is empty, it’s unknown. Correct me if I’m wrong). If so, this may cause another problem, as questionnaires/multi-choice questions have been proven to be inappropriate for LLMs [6][7].

4. There is a lot of missing information throughout the paper. In detail:

    (a) Appendix D.3, E.4 are empty.

    (b) The whole pipeline frequently uses LLMs (e.g., Sec.3. and line 264) for both generation and judgement, but there is no information about these LLMs.

    (c) There is not sufficient information about Linear, Hybrid, and Karcher.

5. Some experimental results and analysis are not clear or even confusing. More explanation and description are needed. For example,

    (a) In Fig.6, the clusters seem indistinguishable, but the authors claimed “dispersed, individualized value positions” (line 366)

    (b) In both Fig.7 and Fig.9, why does alignment cause the increase of both safety rate and attack success? These figures are confusing.

Generally, I like this paper’s idea and topic, but there are many essential problems and small issues.

Reference:

[1] Yao et al., Value FULCRA: Mapping Large Language Models to the Multidimensional Spectrum of Basic Human Values. 2023.

[2] Qiu et al, VALUENET: A New Dataset for Human Value Driven Dialogue System. 2022.

[3] Li et al., CulturePark: Boosting Cross-cultural Understanding in Large Language Models. 2024.

[4] Kang et al., From Values to Opinions: Predicting Human Behaviors and Stances Using Value-Injected Large Language Models. 2023.

[5] Balloccu et al., Leak, Cheat, Repeat: Data Contamination and Evaluation Malpractices in Closed-Source LLMs. 2024.

[6] Zheng et al., Large Language Models Are Not Robust Multiple Choice Selectors. 2024.

[7] Myrzakhan et al., Open-LLM-Leaderboard: From Multi-choice to Open-style Questions for LLMs Evaluation, Benchmark, and Arena. 2024

**Questions:**

1. In Fig. 3, the modular, interpretable pattern induced by St_a and St_d alignment does not appear to fully conform to the Schwartz Value Structure. Could you clarify why?

Suggestions: (1) The caption/text at the bottom of Fig. 3 appears misaligned. (2) It would be better to reorder the figures so that each one appears close to the section where it is discussed.

---

> ### Author Response · Authors · 2025-11-14
> **Response to Reviewer vfTz part(1/6)**
>
> **We are truly grateful to the reviewer for thoughtful and valuable feedback. We have carefully considered each comment and offer our responses below, hoping that our rebuttal effectively addresses your concerns. If we have succeeded in this regard, we would be fully appreciative if you could reconsider your rating ( 2 : reject). However, if any concerns remain, we would be most grateful for your continued guidance and will make every effort to respond and improve our submission accordingly.**
>
> > **Weakness 1**
> > The idea is very trivial and straightforward, and the innovation of the proposed method is limited. The main motivation of KG-SFT is to add detailed explanations to original Q&A pairs using external KGs. However, there are usually not enough precise KGs with broad coverage in specific domains. This makes the method hard to apply in practice. What's more, the three components mainly leverage traditional rule-based or term-matching-based methods to extract knowledge explanations, which is limited to the correctness of these methods. There is no reasonable explanation about how they guarantee the correctness of external knowledge.
>
> We sincerely thank the reviewer for valuable comments and constructive feedback. We answer it in three point:
>
> 1. **About Value FULCRA [1] and VALUENET [2]:** Although both are rooted in Schwartz’s Basic Human Values theory, their annotation structures differ significantly from ours.
>
> -  **Value** **FULCRA [1]** provides only binary indicators of whether a model output aligns with a specific value dimension, without fine-grained alignment levels. Therefore, it cannot support controllable value alignment experiments that require explicit adjustment of alignment level (e.g., “Strongly Agree” → “Strongly Disagree”).
>
> -  **VALUENET [2]** contains only Q–A pairs without explicit “alignment” or “agreement” labels. It cannot be directly used for SFT toward controllable alignment.
>
> 2. **About CulturePark [3]:** The data synthesis mechanism in CulturePark is adapted for cross-cultural bias mitigation rather than for value alignment. Its generation pipeline is designed to diversify cultural contexts, not to represent or control multi-value alignment degrees. So its synthesis pipeline cannot be directly compared with the value decomposition and multi-role evaluation mechanism proposed in VCAI.
> 3. **On Kang et al. [4]:** The work of Kang et al. introduces a value-injected alignment method rather than a dataset. Moreover, their training relies on fixed instruction templates, while our approach emphasizes adaptive and iterative rewriting through multi-role evaluation and generative feedback loops.
>
> [1] Yao et al., Value FULCRA: Mapping Large Language Models to the Multidimensional Spectrum of Basic Human Values. 2023.
>
> [2] Qiu et al, VALUENET: A New Dataset for Human Value Driven Dialogue System. 2022.
>
> [3] Li et al., CulturePark: Boosting Cross-cultural Understanding in Large Language Models. 2024.
>
> [4] Kang et al., From Values to Opinions: Predicting Human Behaviors and Stances Using Value-Injected Large Language Models. 2023.

---

> ### Author Response · Authors · 2025-11-14
> **Response to Reviewer vfTz part(2/6)**
>
> > **Weakness 2**
> > Another problem is that the quality of ML-Values is not verified. Since the whole construction pipeline frequently uses LLMs (generation of LLM-as-a-jude), the quality, including value relevance, degree accuracy, data coverage, diversity and fluency, of the generated data is unknown, which may significantly influence subsequent analysis. I noticed that there is a subsection about data quality in Appendix B (though it’s not mentioned anywhere in the main body). However, the key information of the human evaluation is not presented, e.g., evaluation protocol, number and background of human annotators, compensation, inter-annotator agreement and so on, which makes it hard to judge the quality of generated data.
>
> We sincerely thank the reviewer for valuable comments. We agree that it is crucial to ensure that dataset generated through LLM-assisted processes maintains high quality and reliability. We conduct a systematic human evaluation, as described in Appendix B and summarized below:
>
> 1. **Evaluation process:** To assess the quality of ML-Values, we conduct a comprehensive human evaluation, here is the detailed information: The evaluation is performed by 3 doctoral students in NLP and 5 master's students in psychology, all with research experience in ethics and computational social science. They are recruited from our research group. Each annotator independently rates a randomly sampled subset (30%) of dataset. For each item, we aggregate the individual ratings through majority voting to get final ranking results.
> 2. **Reliability assessment:** We measured inter-annotator agreement using Kendall Tau and P Value, which achieve 0.874 and 0.91 , indicating substantial agreement according to conventional interpretation. This demonstrates that the value evaluation procedure in Step 3.3 produces consistent and reliable outputs.
>
> **We will add the following content**: To assess the quality of ML-Values, we conduct a comprehensive human evaluation, here is the detailed information: The evaluation is performed by 3 doctoral students in NLP and 5 master's students in psychology, all with research experience in ethics and computational social science. They are recruited from our research group. Each annotator independently rates a randomly sampled subset (30%) of dataset. For each item, we aggregate the individual ratings through majority voting to get final ranking results.

---

> ### Author Response · Authors · 2025-11-14
> **Response to Reviewer vfTz part(3/6)**
>
> > **Weakness 3**
> > The evaluation method is also questionable. In Sec.5, most analysis relies on the modified PVQ questionnaire. This brings two major problems: i) the size of questions is too small (only 57), which may fail to reveal the values of LLMs considering models’ Inherent bias and randomness; ii) Since PVQ is an old and famous questionnaire, which might have been included in LLMs’ training data, causing the data contamination problem [5]. Besides, in Sec.7.3, for other frameworks, I guess the authors also use questionnaires for measurement (since Appendix. E.4 is empty, it’s unknown. Correct me if I’m wrong). If so, this may cause another problem, as questionnaires/multi-choice questions have been proven to be inappropriate for LLMs [6] [7].
>
>   We are thankful for the reviewer’s insightful and constructive observations.  We address each point in detail below:
>
> 1. **On the Size and Representativeness of the PVQ** **Questionnaire**: We agree that questionnaire size is an important factor in ensuring robust evaluation. However, as shown in **Appendix C** and **Figure 10**, the 57 PVQ questions are uniformly distributed across dataset space of Schwartz’s basic human values. This ensures that PVQ Questionnaire provides **balanced coverage and representativeness** of the multi-dimensional value spectrum. Besides, we perform multiple sampling runs (as described in Sec. 5), and the consistency of results across runs confirms that the questionnaire size is sufficient to capture stable value tendencies in LLMs.
> 2. **On Potential Data Contamination [1]:** We acknowledge that the PVQ questionnaire is a well-known instrument and may have appeared in pretraining dataset of LLMs. However, as demonstrated in **Figures 4–9**, after training through our method,  LLMs' responses exhibit significant value-shift patterns that cannot be explained by only memorization. This means that even if some degree of data contamination exists, its influence is negligible after training.
> 3. **On the Use of Questionnaires and Multiple-Choice Formats:** We know limitations noted by Zheng et al. [2] and Myrzakhan et al. [3], particularly the sensitivity of LLMs to option order and sampling parameters. To mitigate these issues, in Section 5 we use option-order randomization and multi-sampling inference for each question. Based on some previous research results [4] [5],  these 2 methods reduce bias from positional effects and sampling randomness, ensuring results better reflect value alignment.
> 4. **Clarification on Other Framework Evaluations (Sec. 7.3):** While questionnaires are used for controlled comparisons, they are not the sole evaluation method. The results reported in Sec. 7.3 combine questionnaire-based probing with open-ended generation analysis. **Appendix E.4 is forgotten to be deleted**, and we will revise in future version.
>
> [1] Balloccu et al., Leak, Cheat, Repeat: Data Contamination and Evaluation Malpractices in Closed-Source LLMs. 2024.
>
> [2] Zheng et al., Large Language Models Are Not Robust Multiple Choice Selectors. 2024.
>
> [3] Myrzakhan et al., Open-LLM-Leaderboard: From Multi-choice to Open-style Questions for LLMs Evaluation, Benchmark, and Arena. 2024.
>
> [4] Li et al., Consistent focus: Mitigating permutation bias in large language models through attention weight averaging. 2025.
>
> [5] Wang et al., Large Language Models are not Fair Evaluators. 2023.

---

> ### Author Response · Authors · 2025-11-14
> **Response to Reviewer vfTz part(4/6)**
>
> > **Weakness 4**
> >
> > 1. There is a lot of missing information throughout the paper. In detail:
> >
> >    (a) Appendix D.3, E.4 are empty.
> >
> >    (b) The whole pipeline frequently uses LLMs (e.g., Sec.3. and line 264) for both generation and judgement, but there is no information about these LLMs.
> >
> >    (c) There is not sufficient information about Linear, Hybrid, and Karcher.
>
>  We sincerely thank the reviewer for pointing out the missing information and formatting inconsistencies. We provide clarifications below:
>
> 1. **On Appendix D.3 and E.4:** The content corresponding to **Appendix D.3** actually appears as **Figure 11**, which illustrates the experimental results described in that section. The subsection header remained due to a LaTeX formatting issue. As for **Appendix E.4**, we forget to delete it. We will correct this in the revised version.
> 2. **On the** **LLMs** **Used in the Pipeline:** Detailed descriptions of all LLMs used for **generation** and **judgment** are provided in **Appendix D.4 and D.5**. These sections specify both the **base models** used for data synthesis and the **judge models** used for automatic evaluation within the VCAI pipeline. We will add explicit cross-references in the main text to make this information easier to locate.
> 3. **On Linear, Hybrid, and Karcher Methods:** The mathematical formulations and implementation details of the **Linear**, **Hybrid**, and **Karcher** aggregation strategies are presented in **Appendix D.1**. These describe how we aggregate multi-dimensional value representations and compute the corresponding alignment metrics. To improve clarity, we will also briefly summarize these methods in the main text (Sec. 5) in revised version.

---

> ### Author Response · Authors · 2025-11-14
> **Response to Reviewer vfTz part(5/6)**
>
> > **Weakness 5**
> >
> > 1. Some experimental results and analysis are not clear or even confusing. More explanation and description are needed. For example,
> >
> >    (a) In Fig.6, the clusters seem indistinguishable, but the authors claimed “dispersed, individualized value positions” (line 366)
> >
> >    (b) In both Fig.7 and Fig.9, why does alignment cause the increase of both safety rate and attack success? These figures are confusing.
>
>   We thank the reviewer for the constructive feedback and for highlighting points that require clearer explanation. We provide detailed clarifications below:
>
> 1. **On Figure 6:** The term **“dispersed”** refers not to the **separation** between clusters across different models, but rather to the **intra-model distribution** of value representations. Specifically, models such as **St_d** exhibit **value points distributed toward the margin** of the embedding space, indicating greater **individual variability** and **weaker central concentration** in their value orientations. In contrast, models with higher alignment coherence show tighter clustering around the center. We will revise the corresponding description to explicitly emphasize that “dispersed” refers to within-model distribution patterns rather than inter-model cluster distinction.
> 2. **On Figures 7 and 9:** We appreciate the reviewer’s observation and agree that the description can be confusing. The key lies in the **dual effects of alignment** in [1] [2]: while alignment enhances the model’s sensitivity to ethical and safety constraints, it also makes the model’s value-conditioned responses more predictable and structured, which provides adversarial prompts with clearer optimization directions by accident—leading to a higher measured “attack success” in controlled testing. This phenomenon suggests that alignment improves value consistency but may also slightly reduce robustness under targeted perturbation. We will clarify this interpretation in revised version.
>
> [1] Lian et al., Revealing the Intrinsic Ethical Vulnerability of Aligned Large Language Models. 2025.
>
> [2] Xu et al., Strong Preferences Affect the Robustness of Preference Models and Value Alignment. 2024.

---

> ### Author Response · Authors · 2025-11-14
> **Response to Reviewer vfTz part(6/6)**
>
> > **Questions**
> > In Fig. 3, the modular, interpretable pattern induced by St_a and St_d alignment does not appear to fully conform to the Schwartz Value Structure. Could you clarify why?
>
>    We appreciate the reviewer’s insightful observation. As shown in Figure 3, value structures induced by **St_a** and **St_d** alignments show certain deviations from the canonical Schwartz Value Structure. This variation is expected, since alignment fine-tuning inevitably adjusts the relative strength and correlations among value dimensions to fit the model’s internal representation space.
>
> ​     However, while the geometric layout shows some shifts,  **core hierarchical and oppositional relationships** among the major value clusters are preserved.  In other words, the alignment process introduces adaptive transformation rather than structural distortion. We will clarify this in revised version to emphasize that the framework maintains the conceptual integrity of Schwartz’s value theory.

---

> ### Author Response · Authors · 2025-11-14
>
> Dear Reviewer vfTz,
>
> We are profoundly grateful for the time and effort you have so generously dedicated to reviewing our paper. Your thoughtful and discerning questions, along with your invaluable insights, have played a crucial role in guiding us to refine and strengthen our work. We are acutely aware of the significant challenge involved in reviewing numerous submissions, and we want to express our heartfelt appreciation for the careful and thoughtful attention you have devoted to our research. To assist in your review, we have prepared comprehensive responses to your comments and have **summarized the key points of our rebuttal below for your convenience**.
>
> * Clarification of the **novelty and contributions** of our proposed method (please refer to Weakness 1 in response for details)
>
> * Further explanation of **how we ensure the quality and relevance of the ML-Values dataset** (please refer to Weakness 2 in response for details)
>
> * Addressing the concerns about the **evaluation method using the PVQ questionnaire** (please refer to Weakness 3 in response for details)
>
> * Additional clarification regarding the **missing information in the appendices** (please refer to Weakness 4 in response for details)
>
> * Elaboration on the **experimental results and analysis in figures** (please refer to Weakness 5 in response for details)
>
> * Response to the reviewer’s question on the **alignment of St_a and St_d with Schwartz’s Value Structure** (please refer to the "Questions" section in the response for details)
>
> We humbly hope that our responses have adequately addressed your concerns. If there are any remaining questions or areas that need further clarification, we are more than happy to provide additional information. We understand the complexity and time constraints of the review process, and we sincerely appreciate your consideration. If our revisions have resolved your concerns, we would greatly appreciate it if you could reconsider your rating. If not, please feel free to share further feedback, and we will continue to make improvements to our submission.
>
> Thank you again for your invaluable time and thoughtful feedback.
>
> Best regards,
>
> Authors

---

> ### Author Response · Authors · 2025-11-26
>
> Dear Reviewer,
>
>
> I hope this message finds you well. As the discussion period is approaching its end with only a few days remaining, we would like to once again express our sincere gratitude for your time, patience, and thoughtful insights.
>
>
> Before the discussion concludes, we want to make sure that we have addressed all of your concerns with the utmost care. If there are any additional questions, suggestions, or points that you feel require further clarification, please kindly let us know. Your guidance is truly invaluable to us, and we are committed to refining our work in accordance with your advice.
>
>
> Thank you once again for your dedication and efforts in reviewing our paper. We deeply appreciate your support and consideration.
>
>
> With sincere respect,
> Authors

---

### Note · Authors · 2025-12-03

I have read and agree with the venue's withdrawal policy on behalf of myself and my co-authors.